# Microrobot Path Planning Based on the Multi-Module DWA Method in Crossing Dense Obstacle Scenario

**DOI:** 10.3390/mi14061181

**Published:** 2023-05-31

**Authors:** Dequan Zeng, Haotian Chen, Yinquan Yu, Yiming Hu, Zhenwen Deng, Peizhi Zhang, Dongfu Xie

**Affiliations:** 1School of Mechanical Electronic and Vehicle Engineering, East China Jiaotong University, Nanchang 330013, China; 3340@ecjtu.edu.cn (D.Z.); 2021038085500064@ecjtu.edu.cn (H.C.);; 2Institute of Computer Application Technology, NORINCO Group, Beijing 100089, China; 3School of Automotive Studies, Tongji University, Shanghai 201804, China; 4Jiangxi Tongling Automotive Technology Co., Ltd., Nanchang 330052, China; 5Nanchang Automotive Institution of Intelligence & New Energy, Nanchang 330052, China

**Keywords:** microrobot, path planning, dynamic window method, hybrid path planning, obstacle avoidance

## Abstract

A hard issue in the field of microrobots is path planning in complicated situations with dense obstacle distribution. Although the Dynamic Window Approach (DWA) is a good obstacle avoidance planning algorithm, it struggles to adapt to complex situations and has a low success rate when planning in densely populated obstacle locations. This paper suggests a multi-module enhanced DWA (MEDWA) obstacle avoidance planning algorithm to address the aforementioned issues. An obstacle-dense area judgment approach is initially presented by combining Mahalanobis distance, Frobenius norm, and covariance matrix on the basis of a multi-obstacle coverage model. Second, MEDWA is a hybrid of enhanced DWA (EDWA) algorithms in non-dense areas with a class of two-dimensional analytic vector field methods developed in dense areas. The vector field methods are used instead of the DWA algorithms with poor planning performance in dense areas, which greatly improves the passing ability of microrobots over dense obstacles. The core of EDWA is to extend the new navigation function by modifying the original evaluation function and dynamically adjusting the weights of the trajectory evaluation function in different modules using the improved immune algorithm (IIA), thus improving the adaptability of the algorithm to different scenarios and achieving trajectory optimization. Finally, two scenarios with different obstacle-dense area locations were constructed to test the proposed method 1000 times, and the performance of the algorithm was verified in terms of step number, trajectory length, heading angle deviation, and path deviation. The findings indicate that the method has a smaller planning deviation and that the length of the trajectory and the number of steps can both be reduced by about 15%. This improves the ability of the microrobot to pass through obstacle-dense areas while successfully preventing the phenomenon of microrobots going around or even colliding with obstacles outside of dense areas.

## 1. Introduction

Microrobots have gained a lot of attention in recent years because of the advancement of science and technology, which has led to their widespread application in engineering, agriculture, the military, and transportation [1,2]. One of these is path planning and obstacle avoidance [3]. While most algorithms are better at planning the collision-free operation path of a microrobot in some simple application scenarios, in real life, especially in complex environments, there are often obstacles, people, or other unpredictable things [4], and it is challenging to obtain the global information of the environment. Hence, to achieve autonomous path planning, microrobots must be more adaptive while taking into account operating efficiency and safety in situations with high obstacle density, difficulty in passing, and a high demand for real-time response.

### 1.1. Literature Review

Depending on how well one understands the environment, route planning can be divided into global path planning and local path planning. The goal of path planning is to compute a continuous collision-free trajectory from the starting position to the end location [5]. Global path planning, which is a path search in a situation where map information is available, primarily consists of the greedy Dijkstra’s algorithm [6], the A* algorithm for heuristic search [7], the particle swarm algorithm [8], the genetic algorithm [9], and the ant colony algorithm [10]. A bidirectional search strategy is the foundation of Singh [11] et al.’s proposed time-optimized A* algorithm, which always chooses the path with the least amount of time and runs. The vertices of convex obstacles are transformed into network nodes in Wu [12] et al.’s improved Dijkstra algorithm, which also discovers the obstacle avoidance trajectory between the beginning and ending points and calculates the shortest path through the cost function. In order to acquire the selection factor for the heuristic function, the literature [13] combined the A* algorithm and the ant colony algorithm, focusing on the estimation of the node cost during the search of the A* algorithm by adding the ant colony algorithm with a reward factor. To determine how to maximize the path quality, Li [14] et al. combined a multi-factor fitness function that took into account path length, path energy consumption, and safety. They then used a heuristic median insertion method to create the initial population. Due to the addition of several variables, this approach can, however, make planning more complicated, and it is challenging to ensure that the algorithm’s solution will be effective.

Local path planning is in-the-moment planning for circumstances where some or all environmental data is unclear. The B spline curve based on curve fitting [15], the artificial potential field strategy [16], which drives motion through a virtual potential field, and the DWA algorithm [17] are popular local path planning algorithms. Li [18] et al. created a two-layer local adjustment strategy that adaptively chooses the trajectory adjustment technique in accordance with the complexity of the environment using the local support property of the B-sample and geometric operations. By adding a gravitational field modulation factor, Rostami [19] et al. proposed an improved artificial potential field algorithm for mobile robot obstacle avoidance. This factor reduces the attraction as a linear function as the robot gets closer to the target control, overriding the local optimum. The aforementioned algorithms take into account the reasonable planning of mobile robots but ignore practical efficiency. A decent path-planning algorithm must be logical, effective, and safe all at once.

Fox [20] et al. propose the DWA algorithm to simulate the trajectory set by combining feasible velocity and angular velocity and scoring them to get the best combination to drive the robot’s motion. The DWA algorithm can take into account the robot’s present condition as well as its physical constraints, environmental restrictions, and other factors. The method can enhance the stability of the mobile robot as it approaches the goal point according to the literature’s [21] provided evaluation function for heading optimization, utilizing the distance function as the target guiding coefficient’s weight. Zhang [22] et al. adjusted the dynamic limitations of the DWA velocity space while taking into account the potential overload of vehicle acceleration during operation. A hybrid method for target points was created by Jin [23] et al., using the DWA algorithm for local planning and the A* algorithm for global path planning. The evaluation functions of all the aforementioned algorithms can plan a better path [24], but they all use fixed weights [25]. Additionally, the distance between the robot and target points and obstacles changes dynamically throughout the motion [26], and the safety distance and driving speed of the robot from obstacles should change in real time according to environmental conditions in areas without obstacles or close to target points, which lacks rationality [27]. As a result, the DWA algorithm and IIA are combined in this paper, and the weights are dynamically altered using IIA in response to environmental changes, giving the hybrid algorithm greater environmental flexibility.

### 1.2. Our Contributions

From the current research status, the existing DWA algorithm has the following problems: (1) There is a dearth of research on smart robots crossing dense areas of obstacles as well as a lack of studies on obstacle avoidance planning for robots in complicated environments. Nonetheless, there are instances where driving safety is impacted when robots pass through a lot of obstacles in real life; (2) the traditional DWA algorithm’s current evaluation function is flawed, which causes the irrationality of robot movement close to crossing obstacles; (3) DWA differs from other conventional planning algorithms in that its performance is highly reliant on the selection of the proper weight factors. Usually, our selection of parameters is only based on empirical values, and the DWA algorithm with fixed weights has certain limitations. Although obstacle avoidance is achievable, its effects are general, and it is easy to run into issues with robot avoidance that result in detours or failed obstacle avoidance, which lowers the algorithm’s success rate.

The planning method outlined in this paper is intended to address the challenge of robot obstacle avoidance in circumstances where both dense and sparse areas of obstacles exist. It primarily entails identifying dense areas of obstacles and planning robot mobility through these environments. Optimizing the adaptation for DWA in complicated circumstances uses the immune algorithm with a high diversity preservation method. The method proposed in this research is shown to look for a more perfect trajectory with a lower path length and smaller path and heading variation through 1000 simulation experiments. By using this strategy, non-essential robot detours and collisions can be decreased, and the viability of robot pathways and the success rate of obstacle avoidance can both be boosted. The following are the primary contributions to this paper:By combining the Mahalanobis distance, Frobenius norm, and covariance matrix, a method for judging obstacle-dense areas has been proposed for the first time. The method can locate these areas roughly and help avoid obstacles by estimating the distribution of obstacles overall. This method has the foresight to ensure the safety of robot operation;To improve the ability of the robot to navigate to target points, the original evaluation function of DWA is modified and a new evaluation function based on target points is added;This paper combines EDWA with a two-dimensional analytic vector field method with a good obstacle avoidance effect to produce a multi-module hybrid algorithm to detect the location of obstacle-dense areas in real time and change the planning strategy. DWA’s poor obstacle avoidance effect in dense areas is addressed by this paper by combining EDWA with this method;An improved immune algorithm is created to get the best weight solution based on the algorithm’s convergence iteration and to realize the dynamic change of weight combinations, improving the logicalness of path planning.

In conclusion, the paper is set up as follows: The fundamental theory of the immune algorithm and the traditional DWA algorithm will be introduced in Section 2. The suggested IIA algorithm, as well as the way of judging obstacle-dense areas, the definition of obstacle-dense areas, the critical moments of obstacle-dense areas, and the MEDWA-based planning method, will all be covered in Section 3 of this paper. To demonstrate the viability of the proposed planning method, simulation results are provided in Section 4. The whole content is summarized in Section 5, along with thoughts for further investigations.

## 2. Basic Theory Algorithm

### 2.1. Principle of Traditional DWA Algorithm

The fundamental idea of the traditional DWA algorithm (TDWA) is to limit the sampling speed in consideration of the dynamics, kinematics, and safety of the robot while converting the position control of the robot into velocity control [28,29,30].

The kinematic constraint *V_k_*, where the robot is constrained by the upper and lower limits of its own linear velocity *v* and angular velocity *w*, considering only the motor performance:(1)$$V_{k}=\{(v,w)|v_{min}\leq v\leq v_{max},w_{min}\leq w\leq w_{max}\}$$

The kinetic constraint *V_d_*, the robot acceleration has a limited range, retaining only the maximum acceleration or maximum deceleration that can be reached in a certain time:(2)$$V_{d}=\{(v,w)|v\in [v_{c}-{\dot{v}}_{d}dt,v_{c}+{\dot{v}}_{a}dt],w\in [w_{c}-{\dot{w}}_{d}dt,w_{c}+{\dot{w}}_{a}dt]\}$$
where *v_c_* and *w_c_* are the current linear velocity and current angular velocity, ${\dot{v}}_{m}$ is the corresponding maximum linear acceleration, ${\dot{w}}_{m}$ is the corresponding maximum angular acceleration, and dt is the time interval.

The safety constraint *V_o_*, based on the safety consideration of the robot, states that in order to be able to not collide with the obstacle after braking, the following conditions must be satisfied under the condition of maximum deceleration, where *dist*(*v*,*w*) is the distance of the current trajectory from the nearest obstacle:(3)$$V_{o}=\{(v,w)|v\leq \sqrt{2dist(v,w){\dot{v}}_{m}},w\leq \sqrt{2dist(v,w){\dot{w}}_{m}}\}$$

The smaller velocity space *V_r_* is generated from the above three velocities:(4)$$V_{r}=V_{k}\cap V_{d}\cap V_{o}$$

Finally, the sampling window is determined by the evaluation function of Equation (5):(5)G(v,w)=δ[α⋅heading(v,w)+β⋅dist(v,w)+γ⋅v(v,w)]
where *heading*(*v*,*w*) is the heading angle function, which is used to measure the degree of heading deviation of the predicted end trajectory point. *dist*(*v*,*w*) is the safety evaluation function, which is used to measure the distance between the predicted end trajectory point and the nearest obstacle. *v*(*v*,*w*) is the speed magnitude evaluation function, which is used to measure the driving speed. *α*, *β*, and *γ* denote the weights of the three subevaluation functions, respectively, and *δ* denotes normalization.

### 2.2. Principle of Immune Algorithm

Immunity algorithm (IA) is a type of biological intelligence optimization algorithm created artificially by copying biological immunity mechanisms and fusing them with genes’ evolutionary mechanisms. It models the central concept of how the biological immune system processes antigens, and similarly to conventional optimization algorithms, operators carry out IA’s evolutionary optimization seeking process. Affinity evaluation, incentive evaluation, cloning evaluation, variation evaluation, and other operators are examples of IA operators. The key phases in solving particular issues with IA generally are: (1) defining antigens; (2) initializing the population; (3) calculating affinity; (4) assessing antibody concentration and incentive; and (5) carrying out vaccination operations; and (6) refreshing the population. 

## 3. MEDWA-Based Planning Approach

### 3.1. Judgment Method of Dense Obstacle Area

The typical approach to solving robots via dense obstacles is to improve the current algorithm to increase its stability and convergence, i.e., to improve the algorithm’s performance without taking the obstacles themselves into account. Thus, the algorithm and the dense obstacles themselves are taken into account jointly in this paper, and a method for identifying the dense obstacle area is first provided by combining the Mahalanobis distance, Frobenius norm, and covariance matrix.

#### 3.1.1. Definition of Dense Obstacle Areas

A general definition of an obstacle environment is an area where more than three obstacles are clustered together and the position and size of the obstacles have a direct impact on the passage of a robot. The position of obstacles is determined by their *X* and *Y* coordinates in the model coordinate system, and the volume of obstacles is used to determine the size of obstacles in order to make it easier to quantify the difficulty of a robot’s passage. When there are many obstacles, the volume of data increases dramatically, and the quantity of each obstacle indicator also increases significantly. As a result, determining the correlation between the three types of data for each obstacle is all that is required. The data for the three dimensions, *X* coordinate, *Y* coordinate, and volume, are then used to construct the covariance matrix, which can then be used to determine whether there is a positive correlation, negative correlation, or independent relationship between the three dimensions of the Equation (6).
(6)$$Cov_{ob}=\left[\begin{array}{ccc} k_{x}^{2} & k_{xy} & k_{xv}\\ k_{yx} & k_{y}^{2} & k_{yv}\\ k_{vx} & k_{vy} & k_{v}^{2} \end{array}\right]$$
where element *k_yx_* is the correlation between the *X* and *Y* coordinates of the obstacle, *k_vx_* is the correlation between the *X* coordinates and volume of the obstacle, and *k_vy_* is the correlation between the volume and *Y* coordinates of the obstacle. The stronger the correlation, the lower the element value must be to indicate a negative correlation, and the higher the element value to indicate a positive correlation. Nearly independent elements are those with values close to zero. The diagonal elements are variations in the three dimensions, which we may ignore in this context.

In the obstacle environment depicted in Figure 1, the obstacle area correlates positively. If the robot attempts to pass through the obstacle area, however, due to the unknown nature of the environment, it may cause the robot to “lock” or even collide. As a result, the robot will choose to go around when passing because the environment does not constitute an obstacle-dense area. The obstacle area in the environment depicted in Figure 2 exhibits a negative correlation, making it easier for the robot to move through. The environment does not also represent an obstacle-dense area. From the above two cases, it can be seen that at least one of *k_yx_*, *k_vx_*, and *k_vy_* is greater than zero or less than zero. In the environment shown in Figure 3, the obstacle area is independent, and the robot does not choose to go around and can safely pass the obstacle area, where the values of *k_yx_*, *k_vx_*, and *k_vy_* are close to zero. From the above analysis, it can be obtained that after calculating the covariance matrix of the obstacles in this area, when the *k_yx_*, *k_vx_*, and *k_vy_* values satisfy Equation (7) (according to the nature of the covariance matrix, it is known that *k_yx_* = *k_xy_*, *k_vx_* = *k_xv_*, and *k_vy_* = *k_yv_*), define the region composed of these obstacles as the obstacle dense area.
(7)$$\left\{\begin{array}{l} -0.7<k_{yx}<0.7\\ -1<k_{vx}<1\\ -1<k_{vy}<1 \end{array}\right.$$

#### 3.1.2. Critical Moments in Dense Obstacle Areas

We can simply assess the complexity of the multi-obstacle environment and obtain the covariance matrix of the obstacle-dense area based on the methodology mentioned above. To determine when the robot will visit the area with many obstacles, we must first determine how far away the area is from where the robot is currently located. The calculating approach is illogical and cannot be utilized to accurately assess the degree of closeness between a point and the distribution of points if the distance between the robot and each obstacle is determined separately using the Euclidean distance. As a result, the Mahalanobis distance is described in this paper in order to correct the issue of correlated and inconsistent dimensions in the Euclidean distance. In the scenario of this paper, dense obstacles can be clustered and analyzed as shown in Equation (8), and the Mahalanobis distance from the current position of the robot to the dense area of obstacles is calculated based on the covariance matrix *Cov_ob_*., where *g* is the matrix composed of the current coordinates and volume of the robot, and *μ* is the mean value of the three dimensions of the dense obstacle.
(8)$$M_{a}=\sqrt{{\left(g-\mu \right)}^{T}Cov_{ob}^{-1}(g-\mu )}$$

The calculation allows monitoring in real-time the distance between the robot and the dense area of obstacles, combined with the distance between the dense area and the origin of the coordinates, and taking into account the moment when the robot enters the dense area. Therefore, we can roughly estimate the Frobenius norm, which serves as a measure of the size of a matrix, and the distance between that matrix and the corresponding zero matrices. In layman’s terms, like a point on a two-dimensional plane, the distance from the origin is its Frobenius norm. Assuming that there are *n* (*n* > 3) obstacles in the dense area, as shown in Equation (9), the Frobenius norm ${\left.\left\|Ob\right.\right\|}_{F}$ is calculated from a matrix composed of the position coordinates of the dense obstacles.
(9)$${\left\|Ob\right\|}_{F}=\sqrt{\sum_{X}^{n}\sum_{Y}^{n}{\left|P_{XY}\right|}^{2}}$$

Finally, by combining the Mahalanobis distance and the Frobenius norm to create a new function (10), the instant that the robot enters the obstacle-dense area is defined as when the distance *D* between it and the obstacle-dense area is less than *D_i_*. When *D* between the robot and the densely populated area of obstacles is less than *D_o_*, that is the time at which the robot leaves the densely populated area of obstacles. In this instance, *ε* denotes the Frobenius norm’s additional utility coefficient (0 < *ε* < 1).
(10)$$D=M_{a}/(\varepsilon {\left\|Ob\right\|}_{F})\leq D_{i}$$
(11)$$D=M_{a}/(\varepsilon {\left\|Ob\right\|}_{F})\leq D_{o}$$

### 3.2. MEDWA Algorithm

The heading angle function and obstacle function are optimized in this research and used as the first modules of MEDWA before the obstacle-dense area due to the limitations of the original evaluation function of TDWA. This paper adopts the feedback approach of a single robot proposed in the literature [31] in place of TDWA as the second module of MEDWA to complete the obstacle avoidance planning of robots in obstacle-dense areas and address the issue of the poor obstacle avoidance capabilities of TDWA in obstacle-dense areas. The strategy is based on a class of two-dimensional resolved vector fields with high conflict dissipation and collision avoidance capabilities, merging the attraction vector field of the target point with the repulsion vector field around circular obstacles. This research develops a new target point evaluation function and integrates two prior optimization functions as the third module in order to make the robot approach the target point rapidly after passing through the dense area with obstacles. Finally, IIA generates the weight parameters for the evaluation functions of various modules to achieve dynamic weight adjustment, which boosts the algorithm’s capacity to adapt to changing scene conditions and maximizes its planning effectiveness.

#### 3.2.1. EDWA Algorithm

##### Optimized Heading Angle Function

The objective of optimizing this function is to improve the navigation capability of the robot and the rationality of its planning. In this paper, we consider the case when the robot is far away from the target point (Figure 4) and the case when it is near the target point (Figure 5). Normally, the time of the predicted trajectory is taken as twenty-time steps, and the endmost trajectory point is taken as the evaluation index of the quality of the whole trajectory. Taking the end *Q*_1_ of trajectory A and the end *Q*_2_ of trajectory B as the evaluation points, it is known that the size of the heading angles *θ*_1_ and *θ*_2_ of trajectory A and trajectory B in Figure 4 is similar according to the evaluation of the original heading angle function, which makes it difficult to evaluate the score. In Figure 5, *θ*_1_ > *θ*_2_, the score of trajectory B is higher than that of trajectory A. The planning algorithm chooses the suboptimal solution, which obviously lacks rationality. In this paper, the reference position is changed to the position after a smaller time step on the predicted trajectory, which yields *θ*_4_ > *θ*_3_, and trajectory A will get a higher score.

The above scenario will result from choosing a time step for the predicted trajectory that is too large while choosing a step that is too small will result in a predicted trajectory that is too short and lacks reference. Consequently, a move-out distance *d_m_* should be calculated first, and the time step should be predicted using Equation (12), depending on the current speed *v* of the robot. This will enable the heading angle function *heading*′ (*v*,*w*) to be appropriately optimized.
(12)$$k_{\Delta t}=fix(d_{m}/v)$$
where *fix*(*·*) is the rounding function; the larger the current velocity *v*, the shorter the predicted time step, and vice versa.

##### Optimized Obstacle Function

The goal of optimizing this function is to maximize the planning’s safety while also enhancing the robot’s capacity to avoid obstacles. In this study, we take into account the scenario where the robot is near the target location depicted in Figure 6. The end prediction point *Q*_1_ of trajectory A is located *d*_1_ from the closest obstacle, and the end prediction point *Q*_2_ of trajectory B is located *d*_2_ from the closest obstacle, with *d*_2_ > *d*_1_. It is known that the score of trajectory B is higher than the score of trajectory A according to the evaluation technique of the original obstacle function. The robot will move around the obstruction and lengthen the trajectory and running time if we proceed in accordance with this planning conclusion; hence, route B should be abandoned. Similar to heading angle optimization, just the location following the trajectory beginning point to *d_o_* the time step is taken into account when computing the distance between the robot and the closest obstacle, as seen in distances *d*_3_ and *d*_4_ in Figure 6.

Since the purpose of this function is obstacle avoidance, the value of the time step *d_o_* should be slightly larger than *d_m_* to ensure the safety of the planning. The final optimized obstacle function *dist*′(*v*,*w*) returns the distance between the predicted trajectory and the nearest obstacle.

##### Added Target Point Function

The major objective of including this function is to facilitate the robot’s approach to the target point and reduce the length of the intended trajectory. The goal of the function *target*(*v*,*w*) is to determine the shortest path between the trajectory point at each predicted trajectory’s end and the target point with the weight *ψ*. The two predicted trajectories are contrasted in Figure 7. In the case of the same speed, if we continue with the original evaluation, the end trajectory point *Q*_1_ of trajectory A has a heading angle of *θ*_1_ and the distance to the nearest obstacle is *d*_1_, and the end trajectory point *Q*_2_ of trajectory B has a heading angle of *θ*_2_ and the distance to the nearest obstacle is *d*_2_, clearly indicating that *d*_2_ > *d*_1_, and *θ*_2_ > *θ*_1_. Trajectory A has a higher heading angle score, and trajectory B has a higher obstacle score. It is hard to determine which trajectory is superior because the two trajectories have comparable scores. After incorporating the *target*(*v*,*w*) function, there is *g*_1_ < *g*_2_, and trajectory A will receive a high score to cause the robot to turn at the ideal moment in time.

It is worth noting that the *target*(*v*,*w*) and *heading*′(*v*,*w*) functions both have the ability to navigate to the target point, assisting the robot in approaching the target point from different angles and distances, respectively, but the applicable conditions are different. The trajectory points utilized for scoring are all far from the obstacle and have similar values if the robot is far from the target point. This substantially hinders the *target*(*v*,*w*) function’s ability to make judgment calls while also making the calculation more complicated. *Target*(*v*,*w*) is actually only useful when the target point is close to the robot, especially when there are obstacles nearby. When the target point is far away, *heading*′(*v*,*w*) must still be used to modify the heading. *Target*(*v*,*w*) is thus only added in this paper to the third module of MEDWA, i.e., after the robot has passed through the obstacle-dense area.

Combining the above analysis, the evaluation functions for the different modules in MEDWA were obtained:(13)$$\left\{\begin{array}{l} Module1:G{\left(v,w\right)}_{1}=\delta [\alpha \cdot heading\prime (v,w)+\beta \cdot dist\prime (v,w)+\gamma \cdot v(v,w)]\\ Module3:G{\left(v,w\right)}_{3}=\delta [\alpha \cdot heading\prime (v,w)+\beta \cdot dist\prime (v,w)+\gamma \cdot v(v,w)+\psi \cdot target(v,w)] \end{array}\right.$$

#### 3.2.2. IIA Algorithm

In contrast to other traditional path planning algorithms, TDWA’s performance is highly dependent on the proper set of parameters. Normally, we just choose fixed weights based on experience, but in a complicated environment, it is apparent that this is not always the case. While IA can eventually find the global optimal solution even if it starts with a population of subpar solutions, it does not place as much emphasis on the parameters of the algorithm and the quality of the initial solution.

This paper suggests an IIA method to proactively identify the best weights for MEDWA in scenarios with dense obstacle crossings in order to address the aforementioned issues. To increase the environmental adaptability of the algorithm and achieve the unification of planning in efficiency, rationality, and safety, the number of steps and trajectory length of each completed plan by EDWA are used as the incentive degree of each group of antibodies, and the weights of the evaluation function are dynamically adjusted according to different environmental information. To create the initial antibody set and choose the starting weights for the relevant MEDWA evaluation function, randomly produce *N_p_* = 112 20-bit 01 binary-encoded chromosomes. Path planning is followed by calculating the excitation degree of the current antibody set and repeating the process until all antibodies have been used. Immunization procedures for the antibodies in the top 50% of the incentive degree include immune selection, cloning, mutation, and cloning inhibition. In this research, we present a differential evolution-based mutation operator and an improved cloning operator. The optimal combination of weights (*α**, *β**, *γ**) for MEDWA module I and the optimal combination of weights (*α**, *β**, *γ**, *ψ**) for module III are obtained after Ni immune cycles of training after refreshing the population.

##### Differential Evolution Operator

Differential evolution is a powerful variational operator that is widely used in the field of scientific research [32]. In IIA, differential evolution is used as the variation operation operator in this paper. The variation is achieved by using two different individuals in the population to disturb an existing individual and perform the differential operation. For individuals in the population, the differential evolution of mutant individuals produces the qth generation of mutant individuals according to Equation (14):(14)$$\left\{\begin{array}{l} v_{i}(q)=d_{i}(q)+F(d_{a}(q)-d_{b}(q)),r\leq P_{h}\\ v_{i}(q)=d_{i}(q),r>P_{h} \end{array}\right.$$
where *v_i_*(*q*) is the corresponding variant individual. *d_i_*(*q*), *d_a_*(*q*), and *d_b_*(*q*) are three mutually different individuals randomly selected from the current population and they are also not identical to the target individual. *F* is the variation scaling factor, *r* is the uniform random number in [0,1], and *P_h_* is the variation probability. If a newly generated subindividual exceeds the boundary value, it will be reassigned to the corresponding boundary value.

##### Improved Cloning Operator

The immune selection operator’s choice of specific antibodies will be replicated a predetermined number of times during the cloning operation [28,33]. The most crucial step in the cloning process is determining the number of clones, and dynamically adjusting this number in accordance with the antibody quality can significantly increase the algorithm’s search efficiency. The improved cloning operator is created by the various incentives of the chosen antibodies, as follows:(15)$$N(d_{i})=fix\left(\frac{\lambda f_{inc}(d_{i})}{\sum_{i=1}^{L}f_{inc}(d_{i})}+u\right)$$
where *N*(*d_i_*) is the number of clones of antibody *d_i_*. *λ* is the cloning factor, *L* is the size of the cloned antibody population, and *f_inc_*(*d_i_*) is the excitation degree of antibody *d_i_*. *u* is an integer greater than 1, intended to ensure that each selected antibody will be cloned.

In conclusion, Figure 8 depicts the MEDWA algorithm flow that can be obtained. When compared to the TDWA algorithm, the MEDWA algorithm is better able to identify dense areas of obstacles in complicated situations and uses a vector field method with superior obstacle avoidance capabilities for obstacle avoidance planning. The robot’s navigational performance is enhanced while the path’s length is decreased thanks to the optimized evaluation function in EDWA. To increase the algorithm’s environmental flexibility, the non-dense region can be used by IIA to calculate the ideal weight parameter combination for various situations. With the implementation of the multi-module algorithm, planning efficiency, rationalism, and safety are all taken into account simultaneously [25,34].

## 4. Simulation Validation

The problem of path planning and obstacle avoidance for robots in complicated environments is addressed in this section by combining the MEDWA planning algorithm and designing two scenarios with varying placements of obstacle-dense areas. The proposed algorithm was the subject of 1000 TDWA simulation trials. Eight factors, including step count, path length, deviation from the path, heading angle change, deviation from the heading angle, speed, obstacle distance, and planning success rate, were subjected to comparative analysis.

### 4.1. Simulation Settings

The simulation experiments in this section are run in the environment of Matlab2021a, 2.80 GHz Intel Core i9-10900 with 32 GB RAM of computer configuration. The microrobot is set to move to within 0.25 m of the target point to complete the path planning. The kinematic parameters of the robot are set as shown in Table 1, the parameters of the MEDWA algorithm are shown in Table 2, and the basic settings of the scenario are shown in Table 3. The key parameters of the IIA algorithm are set as follows:*ε* = 0.05; *d_m_* = 0.4 m; *d_o_* = 0.65 m; *N_i_* = 200; *N_p_* = 112; *λ* = 1.5; *u* = 2; *Q*_0_ = 40; *P_h_* = 0.08; *F* = 0.5

### 4.2. Scenario 1

There are 21 circular obstacles with varying degrees of passing difficulty in the environment of scenario 1 where the dense area of obstacles is close to the beginning point, as illustrated in Figure 9. Based on this, experiments are done to verify the performance differences between MEDWA and TDWA with various fixed weights as a robot traverses an area with a dense population of obstacles using path planning simulation.

#### 4.2.1. Simulation Results

##### TDWA Results

TDWA can plan a collision-free path in a complicated environment with numerous obstacles, but it lacks reason and frequently fails to plan. Only a small number of traditional cases are illustrated in this section due to the enormous number of robot collisions that TDWA is responsible for. The path planning outcomes of TDWA failures using fixed weights [1 1 1], [1 1.5 4], and [5 2 2] are shown in Figure 10a–c. The successful path planning outcomes of TDWA with fixed weights [0.3 2 1], [0.6 3 1], and [0.45 1.8 1] are depicted in Figure 10d–f.

The trajectory in Figure 10f is used as a reference. Figure 11a depicts the robot’s changing heading angle as it is being driven. Figure 11b depicts the deviation of the robot’s heading angle, which refers to the departure of this heading from the actual heading of the robot while it is moving, assuming that the line connecting the beginning point and the target point is the ideal heading in an environment free of obstacles. The path deviation below demonstrates the same property. The robot’s path deviation is seen in Figure 11c. Figure 11d depicts the robot’s fluctuating speed. Figure 11e depicts how the robot’s distance from the closest obstacle changes as it is driven.

##### MEDWA Results

A comparison experiment of path planning was undertaken in the same environment to confirm the planning impact of the MEDWA algorithm. Figure 12a depicts the robot’s path as it is being driven. The obstacle-dense area covariance matrix correlation heat map is displayed in Figure 12b. Figure 12c depicts how the robot’s heading angle changes while it is being driven. The deviation in heading angle is shown in Figure 12d. The path deviation is depicted in Figure 12e. The variation in robot speed is depicted in Figure 12f. The variation in the robot’s distance from the closest obstacle during the driving process is depicted in Figure 12g.

In this paper, 1000 TDWA and MEDWA experiments were conducted, and 100 experimental data points from each were randomly chosen. The average values of these 100 experimental data points were then used to calculate the final experimental results, which are displayed in Table 4. Figure 13a,d display the collection of path lengths and step numbers derived from 100 TDWA experiments, respectively. The sets of path length and step number that MEDWA’s 100 experiments produced are depicted in Figure 13b,e, respectively. Figure 13c,f compare the path lengths and step counts obtained from 100 TDWA and MEDWA tests, respectively.

#### 4.2.2. Analysis of Results

The simulation experiments demonstrate that while both algorithms are capable of planning a collision-free path under scenario 1, MEDWA’s planning success rate is up to 99.3%, while TDWA’s is just 11.4%. Robot collisions occur at the deepest portion of the dense obstacle area, as seen in Figure 10a–c, using the majority of the fixed weights. Even if the path can be correctly planned, as shown in Figure 10d,e, the robots will take detours at various inflection points, lengthening the path while causing more severe path deviations. The results of the TDWA algorithm’s planning are depicted in Figure 10f as being somewhat better. The robot makes varying degrees of detours at two obstacle sites close to the target point, as illustrated in Figure 7, as a result of the lack of introduction of the *target*(*v*,*w*) function in the third module. In terms of algorithm performance, MEDWA outperforms TDWA by cutting the intended path length by around 10% and reducing the number of robot running steps by 18.75%. In terms of the quality of the path, the locations where the robot enters and exits the area with a lot of obstacles are indicated by the two different colored “*” shapes in Figure 12a. Although there is no appreciable improvement in the robot’s closest approach distance to the obstacle, the MEDWA-planned path’s heading angle deviation and path deviation do not exceed 0.55 rad and 1 m, respectively. The predicted path deviation for the TDWA is 1.75 m, twice as great as the deviation for the MEDWA, with a heading angle deviation of over 1.2 rad and a speed fluctuation that is significantly more pronounced. No matter the degree of heading angle deviation, deviation of heading angle and path, or speed fluctuation, MEDWA won the test by a wide margin.

### 4.3. Scenario 2

The environment of scenario 2 is an obstacle-dense area close to the target point; as shown in Figure 14, there are 21 circular obstacles with different passing difficulties. The robot has the same starting and ending points that scenario 1 does. Based on this, a simulation experiment is run to determine the performance differences between MEDWA and TDWA with various fixed weights as a robot traverses an area with dense obstacles.

#### 4.3.1. Simulation Results

##### TDWA Results

TDWA can plan a collision-free path in a complicated environment with numerous obstacles, but it lacks reason and frequently fails to plan. Only a small number of traditional cases are illustrated in this section due to the enormous number of robot collisions that TDWA is responsible for. The path planning outcomes of TDWA failures using fixed weights [2.8 1.2 1], [4.5 1 1.5], [1 1 1], [1 2.5 1], [1 5 1], and [2.2 1.2 1] are shown in Figure 15a–f. The successful path planning outcomes of TDWA with fixed weights [0.3 2 1.1], [0.4 2.5 1.8], and [0.45 2 1] are depicted in Figure 15g–i.

The trajectory in Figure 15i is used as a reference. Figure 16a depicts the robot’s changing heading angle as it is being driven. Figure 16b depicts the deviation of the robot’s heading angle. The robot’s path deviation is seen in Figure 16c. Figure 16d depicts the robot’s fluctuating speed. Figure 16e depicts how the robot’s distance from the closest obstacle changes as it is driven.

##### MEDWA Results

A comparison experiment of path planning was undertaken in the same environment to confirm the planning impact of the MEDWA algorithm. Figure 17a depicts the robot’s path as it is being driven. The obstacle-dense area covariance matrix correlation heat map is displayed in Figure 17b. Figure 17c depicts how the robot’s heading angle changes while it is being driven. The deviation in heading angle is shown in Figure 17d. The path deviation is depicted in Figure 17e. The variation in robot speed is depicted in Figure 17f. The variation in the robot’s distance from the closest obstacle during the driving process is depicted in Figure 17g.

In this paper, 1000 TDWA and MEDWA experiments were conducted, and 100 experimental data from each were randomly chosen. The average values of these 100 experimental data were then used to calculate the final experimental results, which are displayed in Table 5. Figure 18a,d display the collection of path lengths and step numbers derived from 100 TDWA experiments, respectively. The sets of path length and step number that MEDWA’s 100 experiments produced are depicted in Figure 18b,e, respectively. Figure 18c,f compare the path lengths and step counts obtained from 100 TDWA and MEDWA tests, respectively.

#### 4.3.2. Analysis of Results

The simulation analysis reveals that while both algorithms are capable of planning a collision-free path under scenario 2, MEDWA’s planning success rate is up to 99.1%, while TDWA’s is just 14.3%. Robots clash inside the dense obstacle area, as seen in Figure 15a–f, employing the majority of the fixed weights. Even if a path can be safely planned, as seen in Figure 15g,h, the robots will take detours at various inflection points, lengthening the path while creating more severe path deviations. The results of the TDWA algorithm’s planning are depicted in Figure 15i as being somewhat better. The robot makes varying degrees of detours at two obstacle sites close to the target point, as illustrated in Figure 7, as a result of the lack of introduction of the *target* (*v*,*w*) function in the third module. In terms of algorithm performance, MEDWA outperforms TDWA by cutting the intended path length by around 10% and reducing the number of robot running steps by 14.63%. In terms of the quality of the path, the locations where the robot enters and exits the area with a lot of obstacles are indicated by the two different colored “*” shapes in Figure 17a. Although there was no significant improvement in the robot’s closest distance to the obstacle, the path heading angle deviation planned by MEDWA was about 0.7 rad, and the path deviation was about 1.5 m. The path heading angle deviation planned by TDWA was about 1.4 rad, twice as large as that of MEDWA, and the path deviation was 1.65 m, and the fluctuation in speed was much larger. No matter the degree of heading angle deviation, deviation of heading angle and path, or speed fluctuation, MEDWA won the test by a wide margin.

## 5. Conclusions

The issue of path planning and obstacle avoidance for microrobots crossing dense obstacles is examined in this paper. A method for judging dense obstacle areas is provided that combines Mahalanobis distance and Frobenius norm to effectively recognize obstacle positions by the microrobot. The evaluation function of the current algorithm is optimized to create a multi-module enhanced DWA algorithm that achieves the multifaceted requirements for effectiveness, direction, and safety in the operation of the microrobot. Additionally, an improved immune algorithm that enhances the system’s ability to adapt to complicated environments achieves the dynamic adjustment of the evaluation function weights. Finally, two scenarios with different locations of dense obstacle areas are created to verify the effects of the planning strategy. The results of the simulation trials demonstrate that the algorithm integrates effectively in a complicated multi-obstacle environment while taking into account the reasonableness and safety of the microrobot operation. As a whole, the planning strategy proposed in this paper increases planning effectiveness and successfully avoids obstacles for the microrobot while maintaining safe operation and minimizing time and resource loss. The technical scheme needs to be further optimized in scenarios that take dynamic obstacles into account, including sensing and recognition, prediction, and planning, as well as strengthening the algorithm’s solution speed. These issues need to be covered in more depth in subsequent research.

## Figures and Tables

**Figure 1 micromachines-14-01181-f001:**
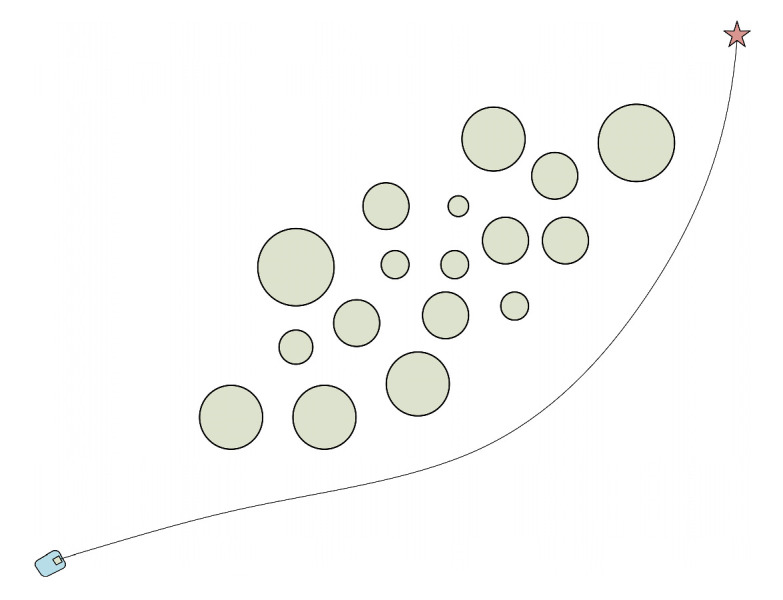
Positive correlation obstacle environment.

**Figure 2 micromachines-14-01181-f002:**
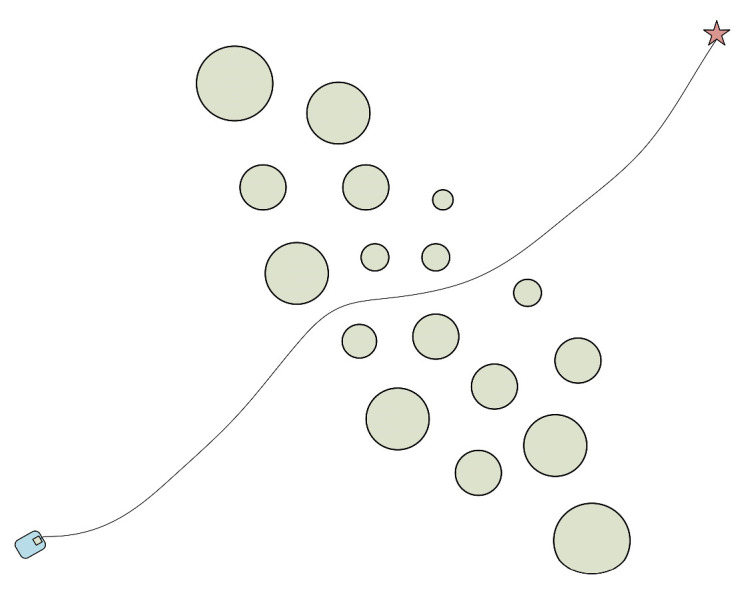
Negative correlation obstacle environment.

**Figure 3 micromachines-14-01181-f003:**
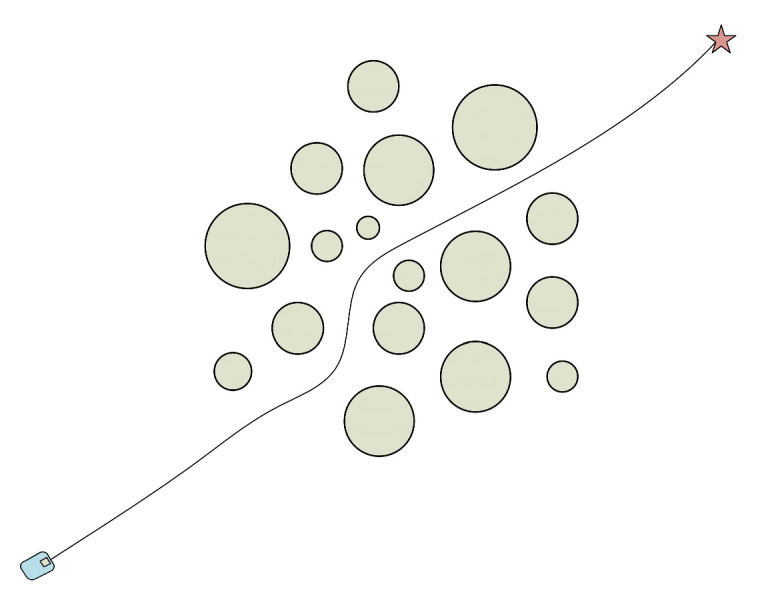
Independent obstacle environment.

**Figure 4 micromachines-14-01181-f004:**
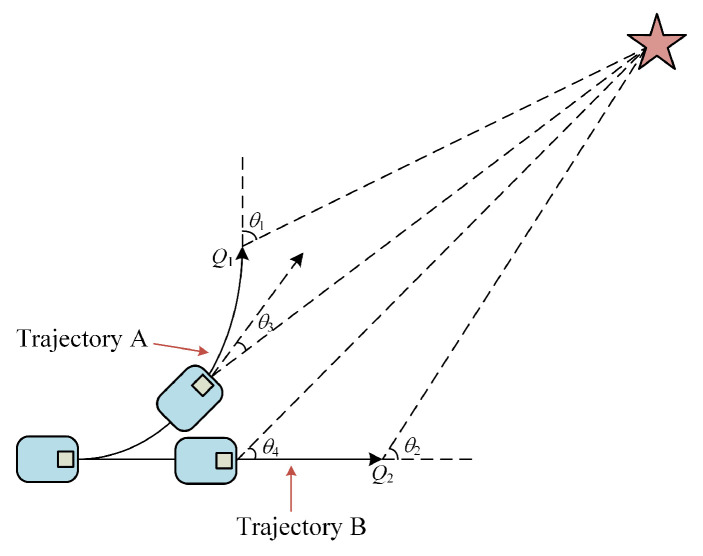
Heading angle planning case 1.

**Figure 5 micromachines-14-01181-f005:**
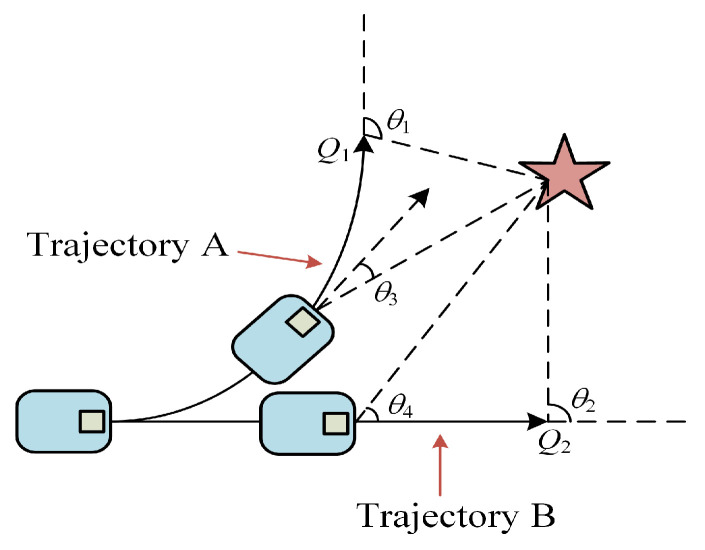
Heading angle planning case 2.

**Figure 6 micromachines-14-01181-f006:**
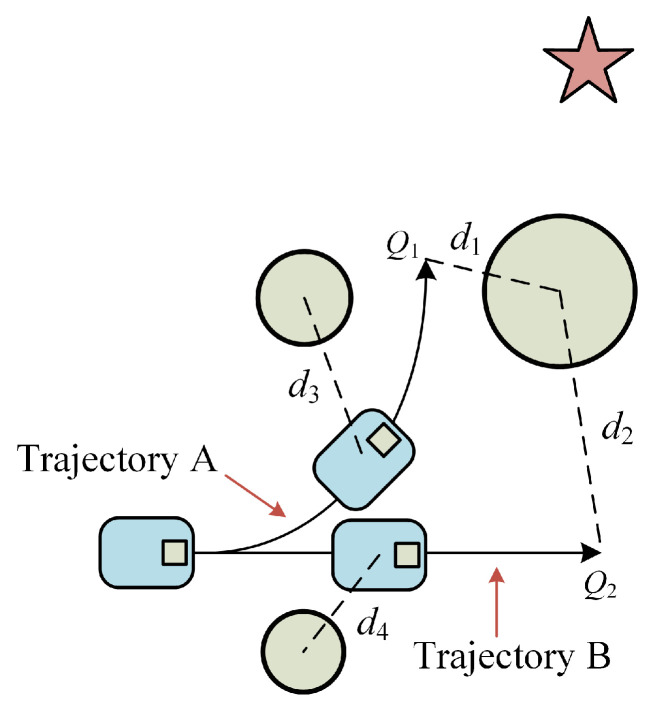
Obstacle planning examples.

**Figure 7 micromachines-14-01181-f007:**
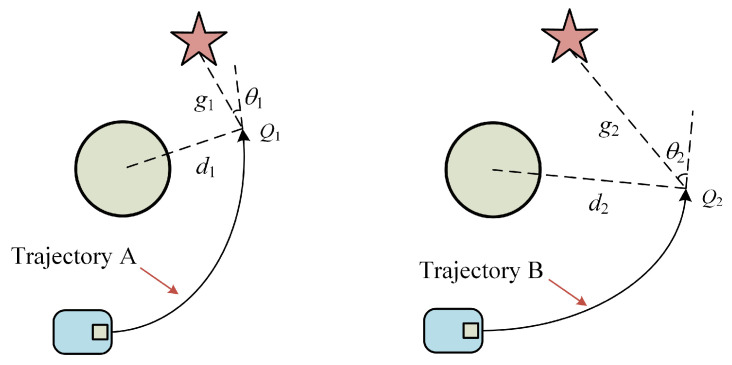
Target Point Planning Case.

**Figure 8 micromachines-14-01181-f008:**
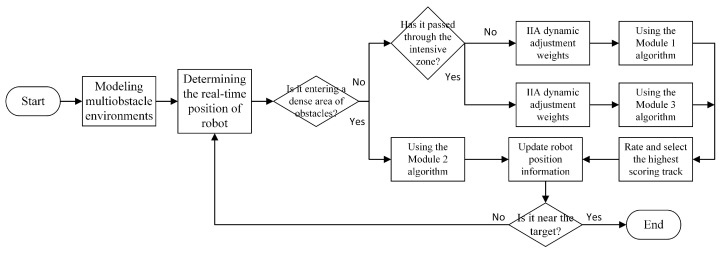
MEDWA algorithm flow.

**Figure 9 micromachines-14-01181-f009:**
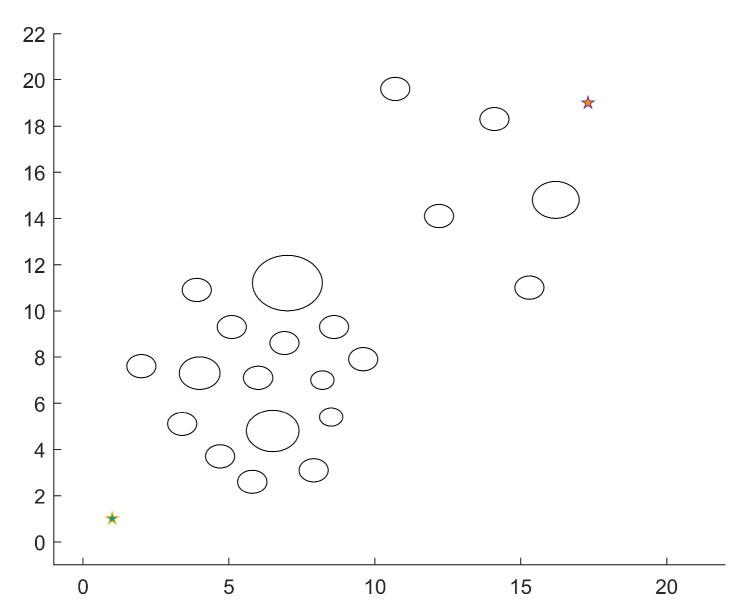
Scenario 1 simulation environment.

**Figure 10 micromachines-14-01181-f010:**
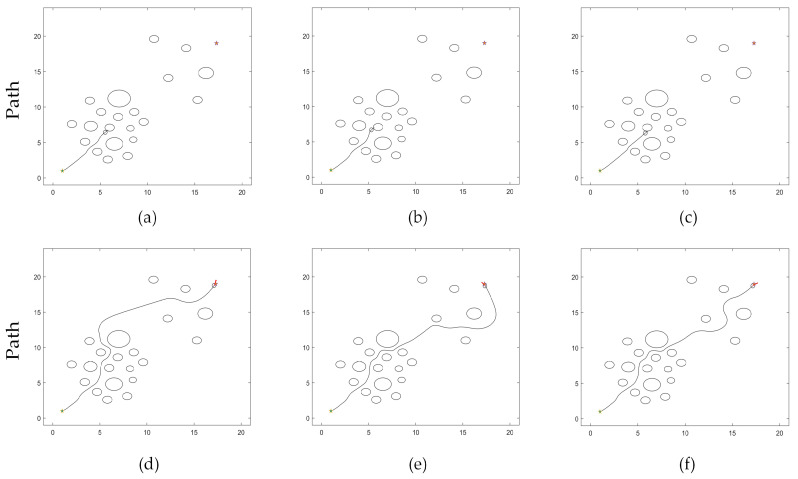
(**a**) Fixed weights [1 1 1]; (**b**) Fixed weights [1 1.5 4]; (**c**) Fixed weights [5 2 2]; (**d**) Fixed weights [0.3 2 1]; (**e**) Fixed weights [0.6 3 1]; (**f**) Fixed weights [0.45 1.8 1].

**Figure 11 micromachines-14-01181-f011:**
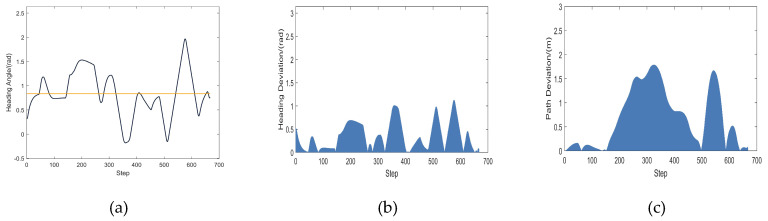
(**a**) Heading angle fluctuation; (**b**) Heading angle deviation; (**c**) Path deviation; (**d**) Speed fluctuation; (**e**) Change in distance to the nearest obstacle.

**Figure 12 micromachines-14-01181-f012:**
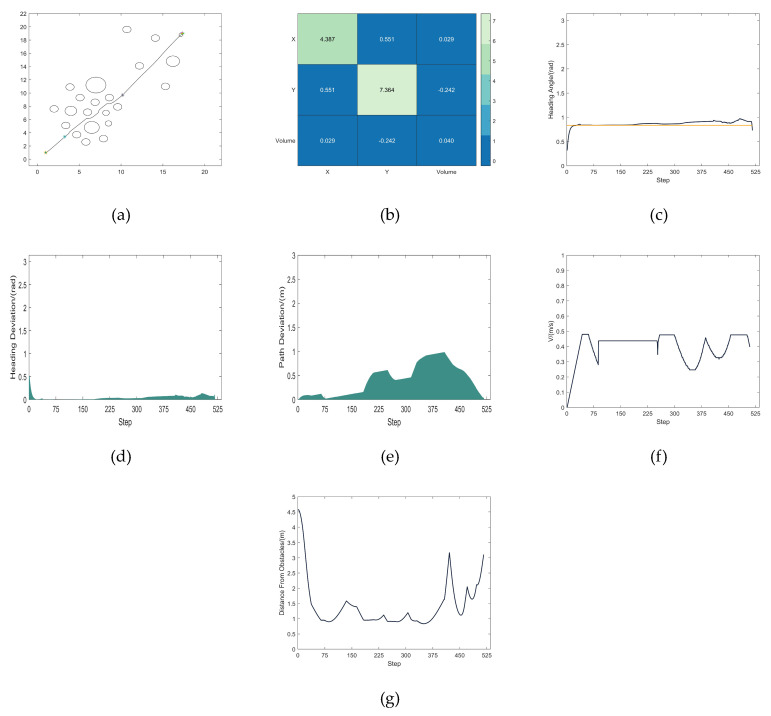
(**a**) MEDWA planning path; (**b**) Heat map of covariance matrix correlation; (**c**) Heading angle fluctuation; (**d**) Heading angle deviation; (**e**) Path deviation; (**f**) Speed fluctuation; (**g**) Change in distance to the nearest obstacle.

**Figure 13 micromachines-14-01181-f013:**
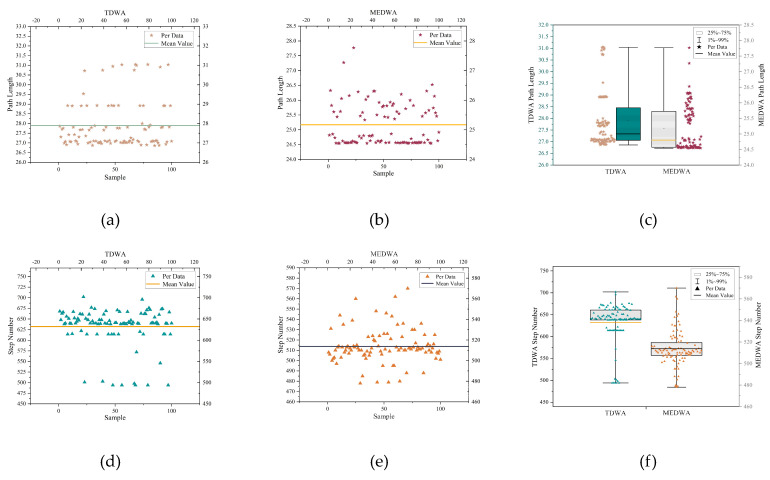
(**a**) TDWA path length set; (**b**) MEDWA path length set; (**c**) Comparison of path lengths between TDWA and MEDWA; (**d**) TDWA step set; (**e**) MEDWA step set; (**f**) Comparison of steps between TDWA and MEDWA.

**Figure 14 micromachines-14-01181-f014:**
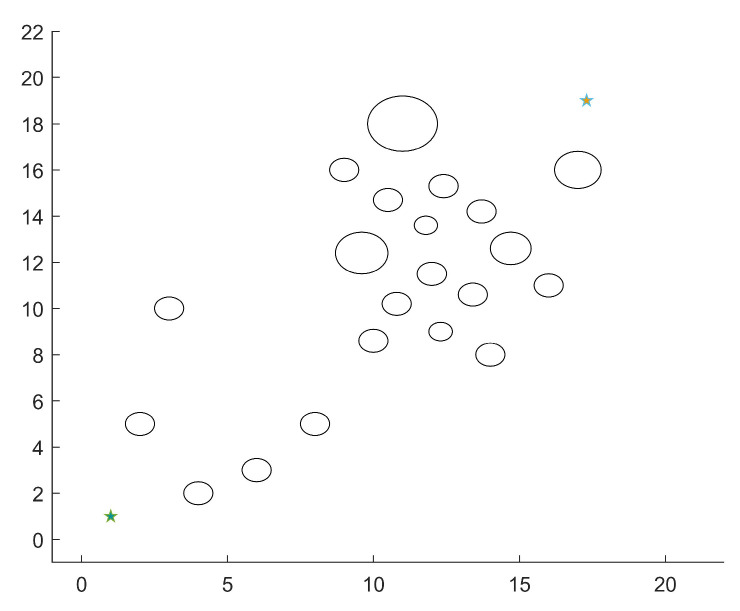
Scenario 2 simulation environment.

**Figure 15 micromachines-14-01181-f015:**
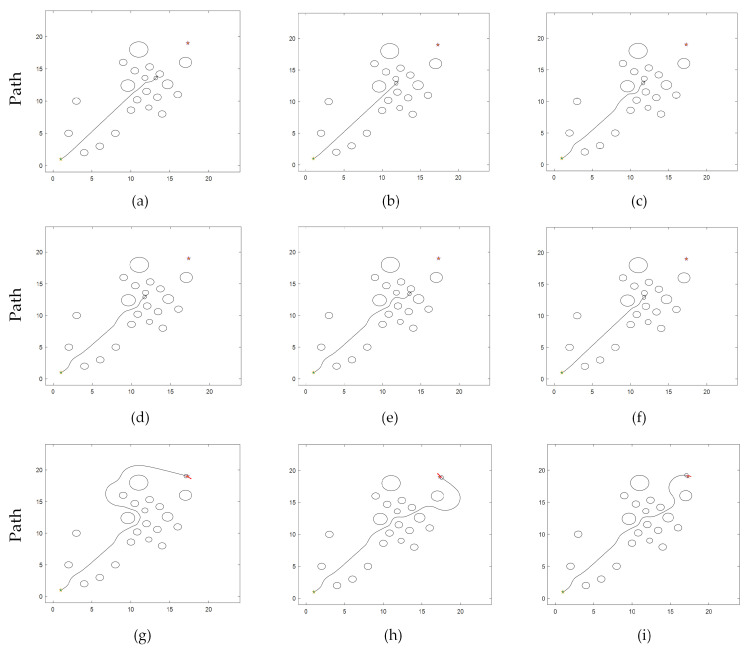
(**a**) Fixed weights [2.8 1.2 1]; (**b**) Fixed weights [4.5 1 1.5]; (**c**) Fixed weights [1 1 1]; (**d**) Fixed weights [1 2.5 1]; (**e**) Fixed weights [1 5 1]; (**f**) Fixed weights [2.2 1.2 1]; (**g**) Fixed weights [0.3 2 1.1]; (**h**) Fixed weights [0.4 2.5 1.8]; (**i**) Fixed weights [0.45 2 1].

**Figure 16 micromachines-14-01181-f016:**
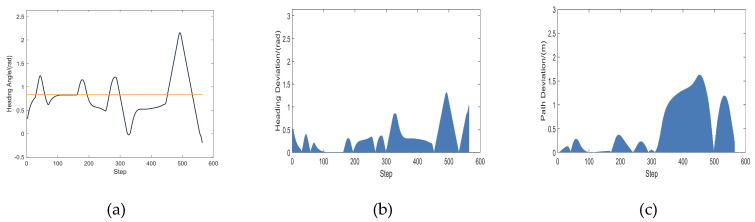
(**a**) Heading angle fluctuation; (**b**) Heading angle deviation; (**c**) Path deviation; (**d**) Speed fluctuation; (**e**) Change in distance to the nearest obstacle.

**Figure 17 micromachines-14-01181-f017:**
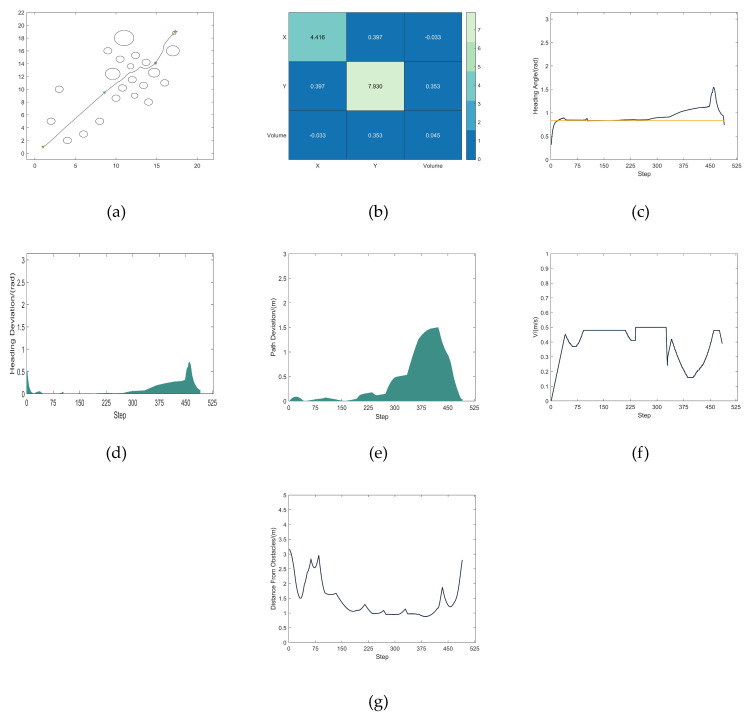
(**a**) MEDWA planning path; (**b**) Heat map of covariance matrix correlation; (**c**) Heading angle fluctuation; (**d**) Heading angle deviation; (**e**) Path deviation; (**f**) Speed fluctuation; (**g**) Change in distance to the nearest obstacle.

**Figure 18 micromachines-14-01181-f018:**
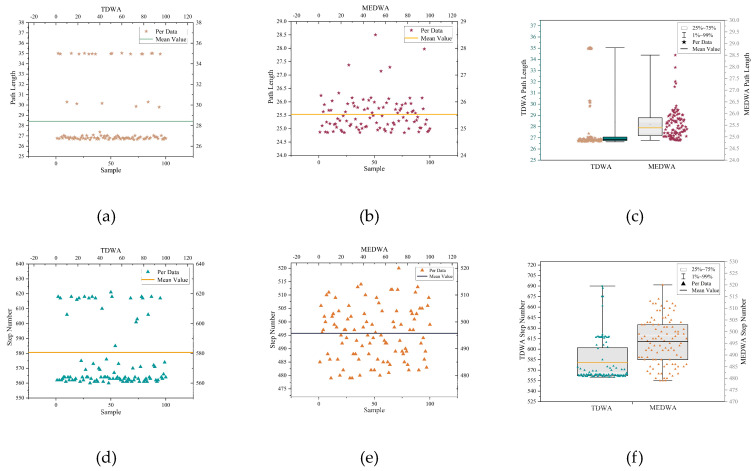
(**a**) TDWA path length set; (**b**) MEDWA path length set; (**c**) Comparison of path lengths between TDWA and MEDWA; (**d**) TDWA step set; (**e**) MEDWA step set; (**f**) Comparison of steps between TDWA and MEDWA.

**Table 1 micromachines-14-01181-t001:** Robot kinematic parameter settings.

Parameter Name	Parameter Value
Minimum linear velocity *v_min_*	0 m/s
Maximum linear velocity *v_max_*	2 m/s
Minimum angular velocity *w_min_*	−π/3 rad/s
Maximum angular velocity *w_max_*	π/3 rad/s
Maximum linear acceleration ${\dot{v}}_{m}$	0.1 m/s^2^
Maximum angular acceleration ${\dot{w}}_{m}$	π/3 rad/s^2^

**Table 2 micromachines-14-01181-t002:** MEDWA algorithm parameters settings.

Parameter Name	Parameter Value
Microbot radius *r*	0.05 m
Linear speed resolution *d_v_*	0.01 m/s
Angular velocity resolution *d_w_*	π rad/s
Time resolution *t_r_*	0.1 s
Trajectory prediction time *t_p_*	3 s

**Table 3 micromachines-14-01181-t003:** Basic scenario settings.

Parameter Name	Parameter Value
Map size	22 m × 22 m
Starting position	(1 m, 1 m)
Target position	(17.3 m, 19 m)
Initial orientation	π/8 rad
Initial velocity	0 m/s
Initial angular velocity	0 rad/s

**Table 4 micromachines-14-01181-t004:** TDWA and MEDWA Comprehensive Comparison.

	Path Length	Step Number	Planning Success Rate
MEDWA	25.172	513.782	99.3%
TDWA	27.895	632.370	11.4%

**Table 5 micromachines-14-01181-t005:** TDWA and MEDWA Comprehensive Comparison.

	Path Length	Step Number	Planning Success Rate
MEDWA	25.533	495.72	99.1%
TDWA	28.411	580.66	14.3%

## Data Availability

The data that support the findings of this study are available from the corresponding author, [Yinquan Yu], upon reasonable request.

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
