# Peer review of "Microrobot Path Planning Based on the Multi-Module DWA Method in Crossing Dense Obstacle Scenario"

_micromachines, 2023, doi:10.3390/mi14061181_

Round 1
Reviewer 1 Report
The authors reported a new path-planning algorithm for microrobots to navigate through areas densely populated with obstacles. This new algorithm is a hybrid between a modified enhanced dynamic window approach (EDWA) for areas with sparse obstacles and a vector field method for areas with dense obstacles. The authors demonstrate the performance of the proposed new algorithm via two simulation scenarios, both having areas with sparse and dense obstacles. Overall, this manuscript presents a new interesting path-planning algorithm.
One issue of this manuscript is that this theoretical work has little relevance to the current experimental work in micro-robotics. The majority of microrobots work in fluidic environments (for potential applications in biomedicine and environments), and their movements are in the low-Reynolds number region where inertial forces and hence accelerations are negligible. Moreover, local velocities and orientations are subject to frequent fluctuations due to thermal noise and hence hard to control precisely, as opposed to the case in macroscopic robots.
Moreover, collisions in fluidic environments in the low-Re number regions are frequent and are not as detrimental to microbots as the collisions with obstacles for macroscopic robots. In other words, microrobots are more collision-tolerant than their macroscopic counterparts. In addition, the obstacles in the fluidic environments are often mobile or can change shapes with the fluidic flow.
These key features of micro-robotics make any control and planning algorithm based on the precise knowledge of the velocity of robots and the exact location of obstacles less applicable to microrobots. Hence, this referee suggests that authors reconsider the context of their work.
Some technical issues are listed below:
1) The algorithm does show promising results in the test simulation environment. How are the computation costs of the new algorithm compared with other similar algorithms?
2) There are many parameters used in the algorithms, as listed in section 4.1. How are they chosen?
3) Would the algorithm show more advantage in three-dimension environments?
4) What is the intuition behind combining the immune algorithm with EDWA?
5) Figures need to be grouped into multi-panel figures.
6) The authors claim that Euclidean distance is computationally more costly than Mahalanobis distance in section 3.1.2. However, if we have precise locations of all the obstacles, the calculation of all the Euclidean distances to a robot can be performed as a matrix operation (vectorized implementation in Python, for example), so it is no less costly than the calculation of Mahalanobis distance.
The writing is decent, but a thorough check on grammar will still be useful.
Reviewer 2 Report
Initially, I congratulate the authors for their research. The manuscript entitled “Microrobot path planning based on multi-module DWA method in crossing dense obstacle scenario” is a good work with several results, written in a direct way. In the opinion of this reviewer, the manuscript seems interesting and worthy of attention. However, there are some aspects that need to be addressed or clarified before further consideration.
Several minor corrections are suggested:
1) The variables in the text must be of the same format as the variables in the equations.
2) The organization of the paper should be included at the end of the Introduction section.
3) The conclusions must be strengthened. The conclusion is to state whether each objective is achieved and what has been achieved. The conclusion part should include any possible impact or benefit of the results obtained.
4) References should be verified and carefully formatting in accordance with the guidelines of the Micromachines.
Reviewer 3 Report
This paper focuses on the problem of path planning and obstacle avoidance for microrobots in dense obstacle environments. The Dynamic Window Approach (DWA), although a good algorithm for obstacle avoidance, struggles in complex situations and densely populated obstacle locations. To address these issues, the paper proposes a multi-module enhanced DWA (MEDWA) algorithm. The MEDWA algorithm introduces an approach for judging obstacle-dense areas by combining Mahalanobis distance, Frobenius norm, and covariance matrix based on a multi-obstacle coverage model. In non-dense areas, the algorithm incorporates enhanced DWA algorithms, while in dense areas, it utilizes two-dimensional analytic vector field methods, which are more effective in planning.
The core of the enhanced DWA (EDWA) algorithm lies in extending the navigation function by modifying the evaluation function and dynamically adjusting the weights of the trajectory evaluation function using the improved immune algorithm (IIA). This improves the adaptability of the algorithm to different scenarios and achieves trajectory optimization. The proposed method is tested in two scenarios with different obstacle-dense area locations. The algorithm's performance is evaluated based on step number, trajectory length, heading angle deviation, and path deviation. The results show that the proposed method reduces planning deviation, trajectory length, and the number of steps by approximately 15%. This enhances the microrobot's ability to navigate through dense obstacle areas while avoiding collisions. Overall, the paper presents a comprehensive approach to path planning and obstacle avoidance for microrobots in complex environments. The proposed MEDWA algorithm demonstrates improved performance in dense obstacle areas and offers potential for further optimization in scenarios involving dynamic obstacles.
While the paper addresses the challenges of path planning and obstacle avoidance for microrobots in dense obstacle environments and proposes the MEDWA algorithm, there are several areas where the paper could be improved:
1. There are a few language mistakes in the manuscript. Please review and make the necessary corrections.
2. The paper mentions that two scenarios were created to test the proposed method, but it lacks sufficient details about the experimental setup, including the specific metrics used for evaluation and any comparisons with other algorithms or baselines. Providing more information about the experimental design and results would strengthen the paper's empirical validation.
3. It would be beneficial for the paper to discuss the limitations of the proposed algorithm and potential avenues for future research. This could include addressing dynamic obstacles, considering real-world constraints, or exploring other optimization techniques.
Round 2
Reviewer 1 Report
The authors have addressed all the issues raised in my report satisfactorily.